# Transmission and Colonization of *Pneumocystis jirovecii*

**DOI:** 10.3390/jof7110979

**Published:** 2021-11-18

**Authors:** Cristian Vera, Zulma Vanessa Rueda

**Affiliations:** 1Grupo de Investigación en Salud Pública, Research Department, Facultad de Medicina, Universidad Pontificia Bolivariana, Medellín 050031, Colombia; 2Department of Medical Microbiology and Infectious Diseases, University of Manitoba, Winnipeg RT3, Colombia; zulmaruedav@gmail.com

**Keywords:** *Pneumocystis jirovecii*, colonization, transmission, natural history, epidemiology

## Abstract

*Pneumocystis* spp. was discovered in 1909 and was classified as a fungus in 1988. The species that infects humans is called *P. jirovecii* and important characteristics of its genome have recently been discovered. Important advances have been made to understand *P. jirovecii*, including aspects of its biology, evolution, lifecycle, and pathogenesis; it is now considered that the main route of transmission is airborne and that the infectious form is the asci (cyst), but it is unclear whether there is transmission by direct contact or droplet spread. On the other hand, *P. jirovecii* has been detected in respiratory secretions of hosts without causing disease, which has been termed asymptomatic carrier status or colonization (frequency in immunocompetent patients: 0–65%, pregnancy: 15.5%, children: 0–100%, HIV-positive patients: 20–69%, cystic fibrosis: 1–22%, and COPD: 16–55%). This article briefly describes the history of its discovery and the nomenclature of *Pneumocystis* spp., recently uncovered characteristics of its genome, and what research has been done on the transmission and colonization of *P. jirovecii*. Based on the literature, the authors of this review propose a hypothetical natural history of *P. jirovecii* infection in humans.

## 1. *Pneumocystis* spp.: Brief Description of Its Discovery

In 1908, at the Experimental Pathology Institute in Rio de Janeiro (Brazil), a physician and researcher, Carlos Chagas (1879–1934), conducted experiments to reproduce trypanosomiasis disease (also called *Chagas*’ *disease)* by injecting trypanosome parasites into rats and guinea pigs. In the histological preparations of the lungs, Chagas visualized some forms of “cysts” that contained eight daughter cells inside them, which had never been described. These forms were misclassified as stages of the lifecycle of *T. cruzi* [1] by Chagas. A year later, a director of the Pasteur Institute located in Sao Paulo, Antonio Carini (1872–1950), sent histological samples to his colleague Felix Mesuil at the Institut Pasteur in Paris. There, Pierre and Mme. Delanoe accidentally discovered a new organism observed by Chagas and Carini when they found “cysts” in the lungs of street rats without any evidence of infection by *Trypanosoma* spp. They suspected that it was a new organism. To confirm their hypothesis, they inoculated these cysts in rats non-infected with trypanosomes, and they did not find any forms of the parasite. Then, this new microorganism was named *Pneumocystis carinii* in honor of Antonio Carini, who provided the samples for the histological study [2,3].

Subsequently, a few published studies on the subject were focused on the distribution of *Pneumocystis* spp. in nature. Between 1913–1917, *P. carinii* was described in rats, mice, guinea pigs, monkeys and rabbits in Brazil, England and Switzerland. Despite its presence in various species of mammal, no evidence of disease was reported in them, which led to diminished scientific interest in this microorganism [4].

It was in 1938 when German researchers Ammich and Benecke described a form of pneumonia of unknown etiology that primarily affected premature and malnourished children, which was later called *plasma cell interstitial pneumonia.* A few years later, thanks to the histological studies of two Dutch doctors: van der Meer and Brug, the association between this pneumonia and *Pneumocystis* was demonstrated for the first time; however, as it all happened during World War II, its discovery went unnoticed [5]. Nevertheless, this pneumonia was of great epidemiological interest in Europe, and particularly in Switzerland, where around 700 cases were reported between 1941–1949, as well as in other countries, such as Czechoslovakia, Italy, Hungary, Germany, Yugoslavia, Austria, Denmark, France, Sweden and Finland, where it had great importance during the 1950s [5]. It was in 1953, when three Czech researchers: Vanek, Jirovec and Lukes, documented that the etiology responsible for this form of pneumonia was *Pneumocystis* spp; as a consequence the discovery is attributed to them [4].

Nonetheless, *Pneumocystis* spp. was still considered a protozoan, due to the presence of the following traits: (i) morphological similarity with its stages and pathogenesis in the host; (ii) absence of typical morphological characteristics of fungi; (iii) no effectiveness of antifungal drugs; (iv) sensitivity to the effect of antiprotozoal agents. Although some investigators argued that the *Pneumocystis* group evidenced some morphological characteristics of fungi [6], it was Edman and Stringer’s studies in the late 80’s that demonstrated, through the analysis of their ribosomal RNA, that the *Pneumocystis* spp. genus belongs to the group of fungi, despite its deficiency of ergosterol in its membrane [7,8]. Figure 1 summarizes the main historical aspects regarding the discovery of *Pneumocystis* spp.

## 2. Nomenclature and Genetic Characteristics

Shortly after the *Pneumocystis* genus was classified in the fungi kingdom, additional data of its DNA showed differences between species of *Pneumocystis*, where its genetic heterogeneity is clearly linked to the species of its host [9]. Based on this, in 1994, under the 3rd International Workshop on Opportunistic Protist (IWOP-3, Cleveland), a temporary use of the trinomial nomenclature was approved (f. sp. “Formae Specialis”) for the name *Pneumocystis carinii* f. sp. *hominis*. in accordance with the requirements of the International Code of Botanical Nomenclature (ICBN) [10]. In 1999, Frenkel et al. named the species that infects humans as *Pneumocystis jirovecii*, in recognition of a Czech parasitologist Otto Jirovec [11]. Other species belonging to the genus *Pneumocystis* have been officially named: *P. murina*, specifically the species that infects mice (*P. wakefieldiae, P. carinii* (rats) and *P. oryctolagi* (rabbits)). Although other *Pneumocystis* organisms have now been found in different mammals, they are currently named using the trinomial system of special forms (*formae speciales*) or other invalid systems. Currently, *Pneumocystis* spp. has the following taxonomic status: Dominio: Eukaria; Kingdom: Fungi; Phylum: Ascomycota; Class: *Pneumocystidomycetes*; Subclass: *Pneumocystidomycetidae*; Order: *Pneumocystidales* and Family: *Pneumocystidaceae* [12].

Thanks to analyses based on the pulsed-field gel electrophoresis techniques (PGFE), the karyotypes of several species of *Pneumocystis* have been obtained, detecting differences in size (6.5–11 Mb) and number of chromosomes depending on the species. Among the most important efforts in the description of *Pneumocystis* spp. genome using electrophoretic techniques, was a study conducted by Cushion et al., who documented the diversity of *P. carinii* karyotypes in 10 types of chronically immunosuppressed rats. In addition, the authors found four distinct karyotypes of *Pneumocystis* spp. comprising a chromosome number between 14 and 16 and an estimated genome size between 7–8 Mb. In the same study, the authors suggested a provisional nomenclature system to differentiate the species of Pneumocystis infecting different types of mammals, including rats, ferrets and humans [13].

Since 1997, the 5th International Workshop on Opportunistic Protists held in Lille, France, and with funding from the National Institutes of Health—NIH—of the United States (from 1999), Drs. Melanie Cushion, George Smulian and James Stinger initiated and led a *Pneumocystis* genome project [14]. In this project, various researchers have made efforts to obtain and sequence the entire genome of *Pneumocystis* spp. by means of the construction of genetic libraries of DNA [15]. This project resulted in the first sequenced and annotated draft of the *P. carinii* genome [16], and subsequently, in 2012, the genome of *P. jirovecii* was documented for the first time, reconstructed with DNA from four bronchoalveolar lavage (BAL) samples from patients with PcP using cell immunoprecipitation and random DNA amplification techniques. An analysis by Cissé et al. documented important genomic characteristics of *P. jirovecii*, including a genome length of 8.1 Mb, a low GC content (29%), and a striking absence of enzymes related to the synthesis of the virulence factors, toxins and enzymes involved in metabolic amino acid pathways, the latter being characteristic of the obligate parasites [17], but it was in 2013 that the complete genomes of *P. jirovecii, P. carinii* and *P. murine* species [18,19] were published. In the same way, in 2016 Ma, L. et al. [20] also assembled and annotated the genomes of *P. jirovecii, P. carinii* and *P. murina* from *P. jirovecii* infected autopsy lung samples from one patient with AIDS, in addition to comparing them with the genomes of *Schizosaccaromyces pombe* and *Taphrina deformans*, free-living saprophytic fungi genetically related to *Pneumocystis* spp. With the findings in the species *P. jirovecii*, we now know that a genome of these species has a length of approximately 8.4 Mb distributed over 20 chromosomes, a GC content of 28.4%, and approximately 3761 protein-coding genes. In this analysis, a gene family with the most significant reduction was observed in the Pfam domains, evidenced by a low number of transporter proteins, transcription factors and enzymes (oxidoreductases, hydrolases, transferases and coenzymes). In addition, other metabolic pathway-related capabilities of the fungus were also found to be reduced and/or absent in all three *Pneumocystis* species analyzed, including *P. jirovecii*, e.g., loss of pathways involved in an amino acid synthesis, (results consistent with those obtained by Cissé in 2006 [17]) and a reduced capacity for nitrogen and sulfur assimilation, lipid and carbohydrate metabolism, glycerol synthesis and cofactor metabolism. On the other hand, Ma, L. et al. also documented enriched protein domains on the cell surface belonging to the Msg superfamily, which have important functions related to the interaction with the host and antigenic variation. These findings could explain the dependence and adaptation of *P. jirovecii* to the human host [20,21]. Likewise, later work by Cissé et al. documented that the adaptation of *Pneumocystis* spp. to the host could be explained, in part, by the highly polymorphic multicopy gene families (msgs), which evolutionarily have provided the fungus with antigenic variation capabilities and necessary mechanisms for immune evasion within the host [22].

On the other hand, Cissé et al. compared the whole genomes of *T. deformans* and *S. pombe* species, applying parsimonious models in order to infer the loss and gain of gene families in the evolutionary history of *Pneumocystis* spp. The results of this study suggested that the common ancestor of *Pneumocystis* spp. may have lost at least 2324 genes, whose functions supplied amino acid and thiamine biosynthesis, nitrogen and inorganic sulfur assimilation, and purine catabolism. Additionally, *P. jirovecii* species displayed a reduction in their arsenal of lytic proteases and machinery for the production of interference RNA, the latter related to genetic functions [23].

Along with the other features, such as the absence of virulence factors that are able to induce cell destruction in the host and difficulty to grow in vitro, it appears that an obligate biotrophy of *P. jirovecii* species is caused by the loss of the essential genes for autonomous subsistence, which makes it dependent on the human lung environment [20,24].

## 3. Transmission of *Pneumocystis jirovecii*

Members of the genus *Pneumocystis* spp. are characterized by being atypical, extracellular, unicellular, low virulence fungi with marked stenoxenism (host specificity) for the lungs of humans and other mammals. Due to the deficiency of a culture system, much of its ultrastructural characteristics and cell biology has been particularly studied from *P. carinii* and *P. oryctolagi* species in animal models, using advanced histochemistry and electron microscopy techniques [25,26], while *P. murina* has been used extensively to study the persistent forms in the lung and immune dynamics of a pneumocystis infection in the host [27,28,29], features that have been extended to other species of *Pneumocystis* [30]. According to available information, it is accepted that a lifecycle of *Pneumocystis* spp. is comprised of two main stages: a mononuclear cell without a cell wall and with a vegetative function, called the trophic form (previously called trophozoite), and a thick—walled form, with multiple nuclear divisions called an ascus form (previously called cystic) [31]. Observing and quantifying the characteristics of its life cycle has not been easy, in part because of the lack of a culture that provides the fungus with the optimal conditions for reproduction. An asexual phase by binary fission and possible endogeny carried out by trophic forms have been insufficiently documented [32,33], and the scientific community is advocating for quantitative studies to understand the life cycle of Pneumocystis spp. [34]. On the other hand, evidence of a sexual phase was first supported by the observation of synaptonemal complexes involved during meiosis in pre-ascus forms of *Pneumocystis* spp. [35]. In addition, recent comparative genomic analyses have elucidated a unique mating-type locus (MAT) composed of three genes involved in cell differentiation and necessary to initiate the mating process and entry into the sexual cycle, suggesting that the reproductive mechanism of *Pneumocystis* species is homothallism [36]. The latter means a single cell of *Pneumocystis* spp. would have the resources necessary to reproduce sexually without the need for another mating type. These findings have generated a new discussion of the reproductive cycle of *Pneumocystis* in the host [34,36,37].

Transmission mechanisms between *Pneumocystis* spp. and its respective hosts are not fully understood [38]. However, studies in the animal model and humans suggest that the most likely route of acquisition of *Pneumocystis* is the airway [39,40,41]. Hughes et al. used a murine model to explore the natural transmission mode of *P. carinii* by subjecting a group of immunosuppressed rats to various sources, including air, water and food contaminated with *P. carinii*. Animals exposed to water and contaminated food were not infected with the fungus, evidencing that the acquisition of *P. carinii* is not possible orally. However, rats exposed to air contaminated with Pneumocystis developed infection, proving that airborne is the most feasible route of transmission from *P. carinii* to the host [41]. Although other studies have documented the presence of *Pneumocystis* genetic material in air samples and water sources [42,43], other authors have found no evidence of *Pneumocystis* DNA in samples obtained outdoors [40,44]. Regarding its infective form, a study conducted in 2010 by Cushion et al. [45] documented that, possibly, an infective stage of *Pneumocystis* is the ascus. In this study, mice were treated with an echinocandin, interrupting the synthesis of β-1,3-D-glucan(BG) and resulting in the depletion of asci but with little effect on other non-BG expressing forms of the fungus, e.g., trophic forms. They then exposed the echinocandin-treated and untreated mice to recipient immunosuppressed mice with the purpose of assessing transmission. They found that mice lacking asci but with viable trophic stages were unable to transmit the airborne infection to uninfected mice. On the other hand, their untreated group could efficiently transmit fungal particles to the recipient mice.

Likewise, the transmission of *Pneumocystis* spp. has also been studied in humans, but the results have not been conclusive. Some authors have reported the presence of *P. jirovecii* DNA in air samples from hospital sites that harbor patients with PcP [46,47,48]. In a study by Bartlett et al., who sought *P. jirovecii* DNA by PCR amplifying the ITS gene in the rRNA region from air samples in places where they had PcP patients, including the rooms where they were hospitalized and their respective residences. In this study, 57% (17/30) and 29% (6/21) of the air samples from hospital rooms and residences, respectively, were positive by PCR [42]. However, the authors argue that the evidence of *P. jirovecii* genetic material in some of the places analyzed does not necessarily indicate the presence of viable or potentially infectious fungal particles [42]; therefore, further studies in molecular epidemiology with more appropriate methodological designs are required, which allow us to describe the possible associations between clinical and molecular variables related to the transmission and acquisition of *Pneumocystis.*

In France, Choukri et al. assessed the airborne dissemination of *P. jirovecii* in 19 immunosuppressed patients with PcP admitted to two hospitals in Paris. They took air samples using the coriolis device and quantified the number of copies/m^3^ of the *P. jirovecii* mtLSU rRNA gene by real-time PCR (qPCR). Air samples were taken at 1, 3, 5 and 8 m away from the bed of each of the patients infected by the fungus. The researchers found *P. jirovecii* DNA in 79% of samples located one meter from patients’ bedrooms, in 69% at 3 m, and 42% and 33.3% at 5 and 8 m, respectively. Additionally, the authors found differences in the distances between the patient and where the air sample was taken, as the number of copies/m^3^ detected was inversely proportional to the distance of the air sample [44].

In the same way, Le Gal et al. conducted a similar survey to the previous one, yet his research question was directed towards the possibility that *P. jirovecii* could be exhaled from patients colonized by this fungus. To answer this, researchers amplified the mtLSU rRNA gene by qPCR from air samples of ten patients hospitalized for various causes and colonized by *P. jirovecii*. For each patient, two air samples were taken for 8 days, 1 and 5 m away from the patient’s bedroom. In addition, five air samples were taken inside and outside the same hospital, and another eight samples in vacant dwellings in the city. They found *P. jirovecii* DNA in 50% (5/10) of the air samples obtained at one meter and 50% (5/10) of these obtained at five meters. Regarding samples from other hospital sites and vacant dwellings, all were negative [37].

Other investigations reported a sudden increase in clusters and outbreaks of PcP, especially in renal, liver and spleen transplant units from various geographic regions [49,50,51]. In Switzerland, Chave et al. described, through a case–control study, a possible outbreak of five PcP cases in renal transplant recipients over a 22-month period. Although pairing between cases and controls was only possible in three patients, the spatiotemporal analysis revealed that transplant patients who developed PcP had more encounters in hospital areas with HIV-AIDS patients who subsequently developed PcP. Although the authors did not assess the presence of colonization in patients with HIV-AIDS, they do suggest the possibility that these have been the source of infection when transmitting Pneumocystis to other immunosuppressed patients [52].

Some studies have applied molecular typing methods to evaluate possible genetic variations involved in the inter-human transmission of *P. jirovecii* [53,54]. A study conducted in France evaluated the possibility of the transmission of *P. jirovecii* inside a hospital in 45 immunocompromised patients with AIDS and kidney transplant patients with a confirmed PcP diagnosis within a period of three years. The authors typed *P. jirovecii* isolates using the multi-target single-strand conformation polymorphism (PCR-SSCP) method, by which they amplified four variable regions of its genome, followed by detection of polymorphisms. Among 45 patients hospitalized with PcP, it was found that eight renal transplant recipients and six AIDS patients had had contact with at least one patient with active PcP in the three months prior to the diagnosis of their own PcP episode. Additionally, in six patients with PcP, the same molecular type corresponding to their respective contact was detected [55].

Cissé O. et al. documented in 2020 an analysis based on the metagenomic techniques to explore the dynamics of human environmental exposure to *Pneumocystis* spp. This analysis was derived from an exposome study [56], where they set out to develop a method for personalized monitoring and tracking of biological and chemical exposure in air; the monitoring included 15 adult participants defined as healthy who were fitted with a personal exposure monitor for the collection of biotic and abiotic samples for aerosol capture with a follow-up period of 890 days at 66 different geographic locations. With the reads obtained and to obtain genetic traces of *P. jirovecii*, the data were compared with reference genomic information available at NCBI, as well as the detection of DNA from different mammals. The authors found evidence of *Pneumocystis* spp. in 24/594 SRA (sequence read archive) files, which represent samples obtained from 4/15 study participants. De novo assembly of the 24 SRA files resulted in 45 contigs with more than 500 nucleotides, of which 37/45 contigs were identified as *P. jirovecii*. Given the host specificity of *Pneumocystis* species and their inability to reproduce outside the host, these results provide evidence of short-range aerosol exposure and possible transmission of *P. jirovecii* and the role of healthy individuals as potential transmitters. Thus, metagenomic methods provide important opportunities for understanding the transmission dynamics of pathogens, in this case, *P. jirovecii.*

According to the epidemiological and molecular evidence available in these studies, the authors do not rule out the possibility that transmission of *P. jirovecii* is possible between infected host–susceptible host in a nosocomial environment. In fact, some important aspects of the possible natural history and route of transmission of this fungus to the human host have been documented. Based on this scientific evidence about the transmission of *P. jirovecii* between infected and susceptible hosts, we propose a hypothetical natural history of *P. jirovecii* infection in humans (Figure 2). However, some important aspects to complete the natural history are unclear: is there a direct contact or a droplet spread? Independently of the immune status of the host, how many infectious particles (asci) would be necessary to develop a PcP or to establish a colonization state?

## 4. Weather and *P. jirovecii*

On the other hand, the association between environmental and climatic factors in the colonization of *Pneumocystis* spp. and PcP has been discussed by several authors [59,60,61]. Demanche et al. studied the frequency of detection of *P. jirovecii* mtLSU rRNA gene by nested PCR in nasal swab samples collected monthly in a closed population of 29 immunocompetent macaques (*Macaca fascicularis*). Although there were no cases of PcP in the animals studied, DNA of the fungus was detected in 34.5% (166/481) of the collected respiratory samples, in which the detection frequencies varied significantly from month to month [62]. Similarly, in Finland, they have studied the seasonal dynamics of *P. carinii* in wild rodent species *Microtus agrestis* and *Sorex araneus* using microscopic detection of asci in respiratory samples, where a high frequency of *P. carinii* was detected in both rodent species in late autumn (November); however, the prevalence was higher in *S. araneus* during all seasons [63].

Considering other environmental factors, a cross-case observational study analyzed the association between temperature, sulfur dioxide and ozone concentrations (O_3_) and the subsequent development of PcP in HIV patients during admission to a medical center in San Francisco, USA, using a conditional logistic regression model with the different time periods. Despite the fact that the authors found that hospital admissions are significantly higher in summer, they found no association to establish environmental risk factors that explain the development of PcP during hospital stay [64]. Similarly, Lubis et al. evaluated monthly variability in PcP incidence through a retrospective cohort study that included 8640 HIV-infected patients seen at hospital institutions in Chelsea and Westminster, England. Among the 792 PcP diagnosed cases, the authors found no significant variations in monthly CD4+T-cell counts (CD4+ TL) or other clinical variables; nonetheless, they found that in January there was a decrease in precipitation levels and, in turn, a slight increase in new PcP cases compared to other months (16% of all new cases). However, the authors do not report biological evidence supporting this possible association [65]. Similar to this finding, the results of Miller et al.’s study found no significant associations between the influence of seasonal variation on PcP mortality, *P. jirovecii* DNA load or the presence of mutations in the DHPS gene [61]. Other studies in Spain agree that PcP occurs more frequently in winter than in any other season of the year [66,67].

Despite the above, studies carried out in other geographical areas have documented conflicting results. In Germany, Sing et al. conveyed a study to analyze the effect of climate on the incidence of PcP between January 1989–December 2006. Amongst the most important variables were changes in average temperature, atmospheric precipitation, along with strength and maximum wind speed. For analysis of the seasonal effect, the stations were classified as: winter (November–February), spring (March–April), summer (May–August) and autumn (September–October). Given that the study includes a period in which there was a significant change in the incidence of PcP in the world (pre-HAART-57.6% and post-HAART 29.6%), the authors concluded that changes in the average temperature (14.2 °C ± 1.2–18.6 ± 1.7 °C) and summer season are associated with an increased incidence of PcP [68]. The differences in the findings reported in the literature make it clear that the epidemiological association between climatic and environmental variables with PcP behavior is still controversial, therefore additional studies that include other variables, such as human behavior, are required, which could have a greater influence on this phenomenon.

## 5. Colonization of *P. jirovecii* in Different Population Groups

Colonization by *P. jirovecii* has been defined as the presence of the fungus in samples of the upper or lower respiratory tract in a host without respiratory signs or symptoms demonstrated by conventional or molecular methods [69,70], In fact, it has been documented that *P. jirovecii* can be a colonizing agent in the respiratory tract of its host. The frequency of colonization by *P. jirovecii* varies according to the type of population and the methods used to detect it. Between the main factors that can favor human colonization are the presence of chronic diseases, immunosuppression, and pregnancy or any form of immune system immaturity (as in newborns) [71]. Nevertheless, any immunocompetent host without any clinical baseline condition can accommodate the fungus asymptomatically in their respiratory tract [72,73]. Therefore, people who are colonized by *P. jirovecii* can be potential infectious sources for transmission to vulnerable people (immunocompromised or infants), as previously described [72,74].

Several researchers in different regions of the world have reported colonization rates by *P. jirovecii* the in adult and immunocompetent population without any respiratory symptoms (between 0–65%) depending on the sample and laboratory technique used for detection [72,73,75,76,77]. Still, an issue about the role that the asymptomatic presence in the airway, as well as the time in life in which the fungus is acquired, remains controversial.

When considering this aspect, two hypotheses were raised about the mechanisms by which this fungus can enter the host. The first considers the de novo acquisition of the fungus. This is based on the findings of different *P. jirovecii* genotypes in each episode of recurrent pneumonia, suggesting that exposure to this pathogen is common [78,79]. The second hypothesis contemplates the asymptomatic acquisition of *P. jirovecii* at some point in life, which probably happens in childhood and is then maintained into adulthood. In fact, it has been suggested that the first contact of the newborn with the fungus occurs at birth, where the mother would be the most likely source of *P. jirovecii* infection [80,81].

Previously, it has been reported that pregnancy is a risk factor for asymptomatic acquisition of *P. jirovecii* [70]. One possible explanation is that in this state several immunological changes occur, as type Th1 cellular response decreases, while Th2 humoral response proportionally increases, besides certain cytokines involved in innate and humoral response, such as: IL-4, IL-6 and IL-10 [82]. A study in Chile found a higher colonization frequency in pregnant than in non-pregnant women by comparing PCR positivity from nasopharyngeal aspirates (15.5 vs. 0%, respectively) [73]. Similarly, in Mexico, another study was carried out to determine the frequency of antibodies against *P. jirovecii* in a group of pregnant women and the frequency with which they transfer these antibodies to their newborn. They found that 48% of the analyzed sera had antibodies against *P. jirovecii*, and evidence of transfer of antibodies in 75% of samples taken from the umbilical cord [43].

In the same way, in children at different ages, antibodies against *P. jirovecii* had also been detected [83,84]. In Spain, it was found that the seroprevalence of *P. jirovecii* increased with age, reaching 52% in children under 6 years, 66% at 10 years and 80% in those aged 13 years [85]. In Chile, they found seroconversion in 85% of healthy children aged between 2–20 months [86]. In fact, a prospective cohort study carried out in Colombia detected *P. jirovecii* at four different times, finding frequencies in mothers and children of 16, 6, 16 and 5% and 28, 43, 42 and 25%, respectively [87]. However, in Peru, a cross-sectional study conducted among pregnant women and newborns documented 5.43% and 0.0%, respectively [88]. Colonization rates vary depending on the type of respiratory sample, laboratory technique, geographic region and infant clinic (Table 1) [73,89,90].

Considering HIV-infected patients, some studies have documented colonization frequencies by *P. jirovecii* between 20–69% detected by PCR in various respiratory samples [92,94]. Leigh et al. reported that colonization by *P. jirovecii* in this group of patients could be influenced by a CD4+ T cell count. The author reported differences between the three groups of HIV patients with CD4 + count greater than 400, 400–60 and less than 60, with colonization frequencies of 10, 20 and 40%, respectively [91]. Pereira et al., on the other hand, describe colonization rates by *P. jirovecii* of 44.8%, more commonly in patients with CD4 +T cell counts less than <200 cells/uL [101]. Nonetheless, other authors have not obtained similar results due to the presence of potentially confounding factors, such as antiretroviral treatment or anti-Pneumocystis prophylaxis, smoking status or geographical location; therefore, the relationship between colonization and CD4 + T cell count is still debated.

Moreover, it has been documented that patients with immune disorders or under immunosuppressive therapy for various reasons are at an increased risk of acquiring asymptomatic *P. jirovecii*. Thus, surveys have been performed in non-HIV immunocompromised patients, in order to describe *P. jirovecii* colonization, where frequencies found in these studies have been far from negligible. Maskell et al. documented a statistical association between glucocorticoid consumption and colonization by *P. jirovecii* in under-aged and immunosuppressed subjects for several reasons, finding a frequency of 44% in patients receiving prednisone at doses higher than 20 mg/day [75]. Another survey in individuals with various autoimmune disorders, found a *P. jirovecii* colonization rate of 16% in sputum samples [100], whereas a Danish study documented a lower percentage. In this research, 367 patients with suspected bacterial pneumonia receiving treatment with corticosteroids, in whom the presence of *P. jirovecii* was demonstrated in 4.4% of subjects, were analyzed by amplification of three loci of its genome [93]. Despite the differences in the frequencies found in various investigations, the results indicate that patients with immune disorders and immunosuppressive states may represent a potential reservoir for *P. jirovecii* circulation within the community.

*P. jirovecii* is also an important agent colonizing the airway in individuals with chronic lung diseases. Concerning this topic, Morris et al. published a review of this issue in 2008, illustrating through 15 articles, that the percentages of settlement in various respiratory samples in this group of patients can vary widely, from 0–100% [103]. Studies in Europe in patients with cystic fibrosis have documented colonization frequencies between 1–22% [99,104] meanwhile, Vidal and colleagues described, in subjects with interstitial lung disease, a *P. jirovecii* colonization rate of 33.8% [95], whereas in a study conducted in China, the percentage rose to 63.3% [102]. Matos et al. explored *P. jirovecii* presence in 45 immunocompetent patients with pathologies in lower respiratory tract samples by conventional molecular methods. In general, the authors found a colonization frequency of 24.4% (11/45) in the entire cohort. Additionally, they highlighted that among patients who had both lung disease and some type of immunosuppressive therapy, the percentage was higher (18%) than in patients who did not receive immunosuppressants (12%) [96].

Subjects with chronic obstructive pulmonary disease (COPD) have a high prevalence of colonization by *P. jirovecii* (16–55%) [76,97,98], and several studies have suggested an important relationship between the presence of *P. jirovecii* and exacerbation of COPD [105]. Nevertheless, this association has not been completely elucidated, and other researchers address their hypotheses to an apparent increase in inflammatory response induced by the fungus in the pathophysiology of COPD due to the increased pulmonary levels of metalloproteinase 12 and the modulation of some immune molecules [98,106]. The presence of *P. jirovecii* in patients with COPD may be related to chronic cigarette smoking, since it has been documented that tobacco is a major risk factor for both outcomes [70,107]. However, studies in non-HIV populations have shown that colonization by *P. jirovecii* represents a risk factor for COPD severity, regardless of cigarette smoking and corticosteroids [97]. The role played by *P. jirovecii* in the COPD population is not fully elucidated, and the issue is currently under investigation.

## 6. Conclusions

According to the above, from the discovery of *Pneumocystis* as an independent genus to the present day, there have been numerous investigations that have been conducted in the fields of basic biology, clinical manifestations, and epidemiology. However, scientific gaps are also evident due to its niche in nature, as well as the mechanisms of transmission and pathogenesis of PcP. In addition, the role that this fungus has as a colonizing agent in the lung of its host and the moment at which it enters the respiratory system, are still topics of current research. Today, the assumptions made about whether the fungus is acquired in the first days or months of life and remains there until its reactivation or on the contrary whether the acquisition of *Pneumocystis* occurs spontaneously at any time have not been completely clarified. In fact, only one study has been conducted in newborns, which evaluated the presence of the fungus in pregnant women and their newborns through molecular techniques with a follow-up of mothers and their offspring to identify when transmission of *P. jirovecii* takes place. Further studies that assess this aspect are needed to understand the precise moment when colonization or infection by the fungus occurs and how it varies over time in order to understand whether it is a temporary agent or whether it remains latently in the lung.

In conclusion, there has been significant progress in the study and understanding of *Pneumocystis jirovecii*’s biology in a human beings; however, there are still many controversies or unknowns in different fields, especially the mechanisms used to enter and leave the host, incubation and transmissibility periods, host factors that enable acquisition, and preventive measures to avoid infection of susceptible individuals. In this sense, conducting basic and epidemiological research about this topic constitutes an essential task the expand our knowledge of the biology and pathogenesis of PcP.

## Figures and Tables

**Figure 1 jof-07-00979-f001:**
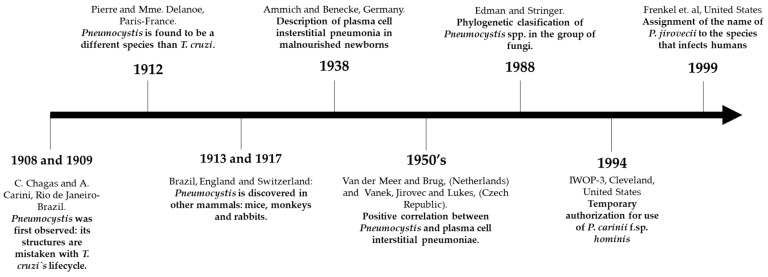
Chronological illustration of the most important events in the discovery of Pneumocystis *jirovecii* from 1908–1999.

**Figure 2 jof-07-00979-f002:**
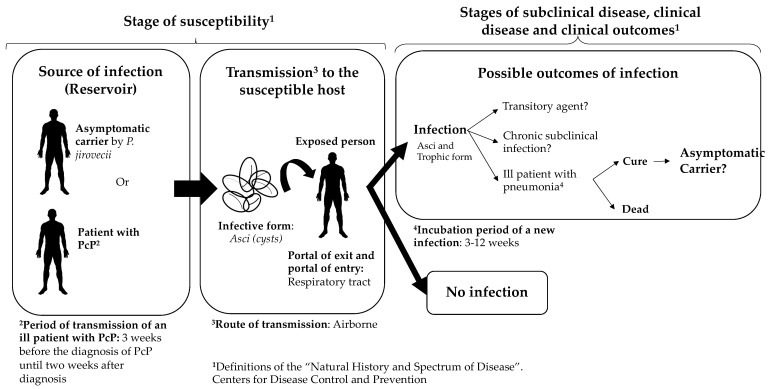
Hypothetical representation of the natural history of *Pneumocystis jirovecii*. Some of the references that support the definitions used in this figure are: route of transmission [38,39,40,41,57,58], infective form [45], incubation period and period of transmission of an ill patient [55]. The question marks in some parts of the figure mean that there are several aspects that are not clear and need further research. We used the definitions of the “natural history and spectrum of disease”, as well as the “chain of infection” in “epidemiology of infectious diseases” from the Centers for Disease Control and Prevention. Based on those definitions, and the literature reviewed, we summarized that the reservoir of *P. jirovecii* is humans and that the transmission is from humans to humans. The source of infection can be asymptomatic carriers or a person with *P. jirovecii* pneumonia. It has been described that there are carriers of *P. jirovecii*, both asymptomatic carriers and people with underlying conditions that are colonized by *P. jirovecii*. However, it is unclear for how long these carriers can be colonized and whether there is a transitory state (it is unclear if the person can be colonized by days, weeks, months, or even years). The portal of entry and exit is the respiratory tract, and the most plausible route of transmission in *P. jirovecii* is airborne.

**Table 1 jof-07-00979-t001:** Documented frequencies of colonization according to different clinical conditions of the host.

Author	Year	Country	Patients	*n*	Colonization Percentage	Biological Sample ^2^	Diagnostic Technique ^3^
Leigh TR [91].	1993	UK	HET ^1^, HOM ^1^ and HIV-seropositive	90	Between 10 and 40%.	IS	PCR
Matos O [92].	2001	Portugal	Adults HIV-infected	104	14.5%	IS and OW	PCR and conventional stain
Helweg-Larsen J [93].	2002	Denmark	Adults with suspected bacterial pneumoniae	367	4.0%	BAL, tracheal aspirates and sputum	PCR
Vargas SL [73]	2003	Chile	Immunocompetent adult, pregnant and non-pregnant women	33 and 28, respectively	15.0% and 0.0%, respectively	Nasopharyngeal aspirates	PCR
Huang L. [94].	2003	EEUU	HIV-infected Adults	32	69%	IS or BAL	PCR
Maskell NA [75].	2003	EEUU	Immunosuppressed HIV-negative adults	93	18.0%	BAL	PCR
Vidal S [95].	2006	Spain	Adults with Interstitial lung disease	80	33.8%	BAL	PCR
Matos O [96].	2006	Portugal	Immunocompetent adults with pulmonary disease and	45	24.4	BAL	PCR
Nevez G [76].	2006	France	Adults with COPD and healthy adults	50 and 30, respectively	16% and 0%, respectively	Sputum	PCR
Morris A [97].	2006	EEUU	Adults with COPD	68	19.1%	Lung tissue Pendiente	PCR
Calderón EJ [98].	2007	Spain	Adults with COPD	51	55.0%	Sputum	PCR
Gal SL [99].	2010	France	Adults and children with cystic fibrosis	76 patients and 146 samples	1.3%	Sputum	PCR
Mekinian A [100].	2011	France	Adults with systemic autoimmune diseases	67	16.0%	IS	PCR
Pereira RM [101].	2014	Brazil	HIV-seropositive Adults	58	44.8%	Oropharyngeal samples	PCR
Wang D-D [102]	2015	China	Adults with chronic pulmonary diseases	98	63.3	Sputum	LAMP and PCR
Vera C [87]	2017	Colombia	Immunocompetent and pregnant women and newborns	43 and 43, respectively	46.5% and 74.4%, respectively	NPS	PCR
García C. [88]	2020	Perú	HIV-negative women and newborns	92 and 87	5.43% and 0.0%, respectively	OW, NPS, and placenta samples	PCR

^1^ HET, healthy heterosexual males; HOM; homosexual males. ^2^ IS, induced sputum; BAL, bronchoalveolar lavage; OW, oral washes; NPS, nasopharyngeal swabs. ^3^ PCR, polymerase chain reaction; LAMP, loop-mediated isothermal amplification.

## Data Availability

Not applicable.

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
