# Peer review of "Transmission and Colonization of Pneumocystis jirovecii"

_jof, 2021, doi:10.3390/jof7110979_

Round 1
Reviewer 1 Report
The co-authors have assumed a noteworthy task to summarize and review the “Transmission and colonization of Pneumocystis jirovecii”. While there are many accurate representations of published articles, there are some inaccuracies that must be addressed prior to publication. This particular field, the study of Pneumocystis species, has suffered from many controversies, inaccurate attempts to name these fungi, incorrect nomenclature, among other problems. Thus, it is incumbent that the authors ensure that their statements are accurate so that inaccuracies are not further promulgated. By citing these problems, I hope the authors take their responsibility to heart. Care must be taken to accurately describe the data and information in the referenced studies.
General: The review needs to be edited. There are many non-English words used and problematic grammar constructs.
- Abstract: what evidence exists the transmission route is (indirectly by aerosols, directly by contact or droplet spread). I would have thought the opposite.
- Section 1: lines 33-35. It would be helpful to the readers how the Delanoes showed that the “cysts” were not part of the trypanosomal life cycle and were a genus and species distinct from those protozoan parasites. (suggested reference: CRC Press, Pneumocystis carinii Vol. 1 Walter T. Hughes, 1987)
- Lines 62-63. The lack of growth in culture is not a discerning attribute for fungi or protozoa. There are members of both that cannot be cultured.
- The citation for IWOP-3 is incorrect ref. 9). The correct citation is: Stringer JR and The Pneumocystis Workshop, Revised Nomenclature for Pneumocystis carinii, J. Euk. Microbiol. 41:5 (121s-122s), 1994.
- The citation, Frenkel et al. 1999 is not the original publication where Frenkel discerned the difference between the rat vs human species. It is: Frenkel JK. 1976. Pneumocystis jirovecisp. from man: morphology, physiology, and immunology in relation to pathology. Natl. Cancer Inst. Monogr. 43: 13-27.
- Embarking on a review of nomenclature is challenging and the authors are referred to the excellent summary of the problematic past in Redhead et al. 2006, their reference 11.
- Line 79: canis and P. macacae have not been formally described according to the ICBN, these names are invalid.
- Line 99: “genetic libraries of mRNA” the reference cited, #14 does not describe such libraries. This citation referred to DNA libraries.
- Line 108: “short” genome is not standard for a reduced genome size.
- Lines 114-121. Biotrophy was first suggested in the following publication: Cushion MT, Smulian AG, Slaven BE, Sesterhenn T, Arnold J, Staben C, Porollo A, Adamczak R, Meller J. Transcriptome of Pneumocystis carinii during fulminate infection: carbohydrate metabolism and the concept of a compatible parasite. PLoS One. 2007 May 9;2(5):e423. doi: 10.1371/journal.pone.0000423. PMID: 17487271; PMCID: PMC1855432.
- Lines 127-132. Pneumocystis murina has been used extensively for immunological studies, evaluation of drugs, and many other applications.
- The authors should update their discussion about the lifecycle by reading recent publications from Hauser, P.
- Line 158: What is intended by the word, “homologues”?
- Figure 2 is difficult to interpret and could be simplified. Also- what is new about this proposed model? Also- again, what is the evidence for droplet spread?
- Lines 291-299: the asymptomatic carriage and subsequent reactivation is no longer thought of as a viable source of active infection. The authors need to delve into the literature on this topic.
- Table 1 needs some kind of organization, perhaps by year of publication. Otherwise, it is hard to follow.
Author Response
Dear Editor,
We want to thank for the detailed and constructive review, including the amazing references and clear suggestions and feedback that the two reviewers did about our paper. We feel priviledge that experts in this field agreed to review our paper and we think this is one of the most educational feedback that we have ever received. We address all your comments and they were very helpful to improve our paper dramatically. We also want to thank that both reviewers gave us the opportunity to improve our paper despite our English mistakes.
We paid the review of the English by a native English speaker, but we haven’t received yet that file. As we already requested one extension because we were waiting for this version, we decided to submit the paper with our changes and corrections to meet the deadline, but we clarify that we are still waiting for the review of our English.
Please find the answers to each comment or question below. Based on your petition, we send to the
Journal of Fungi the article with the changes made "Track changes", so that it can be easily identified.
Reviewer(s)' Comments to Author: Referee: 1
The review needs to be edited. There are many non- English words used and problematic grammar constructs.
Answer: According to the reviewer's recommendation, the language of the manuscript is under review by a native speaker.
- Abstract: what evidence exists the transmission route is (indirectly by aerosols, directly by contact or droplet spread). I would have thought the
Answer: Because we received several comments about this particular topic and one of the figures, we would to share with the reviewers that we are using the epidemiological definition of chain of infection in Epidemiology of infectious diseases (https://www.cdc.gov/csels/dsepd/ss1978/Lesson1/Section10.html#ALT119). There are several different classifications for the routes of transmission, however, we used the CDC definitions. “Route of transmission: Direct (direct contact, droplet spread) and indirect (airborne, vehicle-borne, vector-borne).
In addition, we used the definition of “reservoir of an infectious agent is the habitat in which the agent normally lives, grows, and multiplies. Reservoirs include humans, animals, and the environment. The reservoir may or may not be the source from which an agent is transferred to a host. Human reservoirs. Diseases that are transmitted from person to person without intermediaries. A carrier is a person with inapparent infection who is capable of transmitting the pathogen to others.
Asymptomatic or passive or healthy carriers are those who never experience symptoms despite being infected. Carriers commonly transmit disease because they do not realize they are infected, and consequently take no special precautions to prevent transmission. Symptomatic persons who are aware of their illness, on the other hand, may be less likely to transmit infection because they are either too sick to be out and about, take precautions to reduce transmission, or receive treatment that limits the disease. Portal of exit is the path by which a pathogen leaves its host. The portal of entry refers to the manner in which a pathogen enters a susceptible host.”
Based on these definitions, and the literature reviewed, we summarized that the reservoir of Pneumocystis jirovecii is humans to mention that the transmission is from humans to humans. Several papers have reported this finding (for example: Le Gal S, Damiani C, Rouillé A, Grall A, Tréguer L, Virmaux M, et al. A cluster of Pneumocystis in-fections among renal transplant recipients: molecular evidence of colonized patients as potential infectious sources of Pneumocystis jirovecii. Clin Infect Dis. 2012;54(7):e62-71, Cissé OH, et al. Humans Are Selectively Exposed to Pneumocystis jirovecii. mBio. 2020;11(2)).
In the case of Pneumocystis jirovecii, the portal of exit and entry is the respiratory tract, and the most plausible route of transmission in Pneumocystis jirovecii is airborne (The references 36-41, and 57- 58 support this argument: Cissé OH, et al. Humans Are Selectively Exposed to Pneumocystis jirovecii. mBio. 2020;11(2); Dumoulin A, et al. Transmission of Pneumocystis carinii Disease from Immunocompetent Contacts of Infected Hosts to Susceptible Hosts. Eur J Clin Microbiol Infect Dis. 2000;19(9):671-8; Le Gal S, et al. Pneumocystis jirovecii in the air surrounding patients with Pneumocystis pulmonary colonization. Diagn Microbiol Infect Dis. 2015;82(2):137-42; Hughes WT, Bartley DL, Smith BM. A natural source of infection due to pneumocystis carinii. J Infect Dis. 1983;147(3):595; Gigliotti F, Wright TW. Pneumocystis: where does it live? PLoS Pathog. 2012;8(11):e1003025 and Le Gal S, Damiani C, Rouillé A, Grall A, Tréguer L, Virmaux M, et al. A cluster of Pneumocystis in-fections among renal transplant recipients: molecular evidence of colonized patients as potential infectious sources of Pneumocystis jirovecii. Clin Infect Dis. 2012;54(7):e62-71).
Finally, it has been described that there are carriers of Pneumocystis jirovecii, both healthy carriers or people with underlying conditions that are colonized by P. jirovecii. However, it is unclear for how long these carriers can be colonized and whether there is a transitory state (it is unclear if the person can be colonized by days, weeks, months or even years).
We changed the summary as follows: “Important advances have been made to understand P. jirovecii, including aspects of the biology, evolution, lifecycle, and pathogenesis; it is now considered that the main route of transmission is airborne and that the infectious form is the ascii (cyst), but it is unclear whether there is transmission by direct contact or droplet spread”.
- Section 1: lines 33-35. It would be helpful to the readers how the Delanoes showed that the “cysts” were not part of the trypanosomal life cycle and were a genus and species distinct from those protozoan parasites. (suggested reference: CRC Press, Pneumocystis carinii 1 Walter T. Hughes, 1987)
Answer: Thanks for this suggestion. We could not have access to the text CRC Press, Pneumocystis carinii Vol. 1 Walter T. Hughes, 1987, but we were able to access the book Part One. THE ORGANISM. 1. Historical Overview, Pneumocystis Pneumonia 3rd edition, and two more reviews that summarized Delanoe’s experiments. We included in the paper the following description: “Pierre and Mme. Delanoe discovered accidentally the new organism observed by Chagas and Carini, when they found "cysts" in the lungs of street rats without any evidence of infection by Trypanosoma spp.
They suspected that it was a new organism. To confirm their hypothesis, they inoculated these cysts in rats non-infected with trypanosomes, and they did not find any forms of the parasite. Then, this new microorganism was named Pneumocystis carinii in honor of A. Carini, who provided the samples for histological study [2,3]”.
- Lines 62-63. The lack of growth in culture is not a discerning attribute for fungi or There are members of both that cannot be cultured.
Answer: We agree with the reviewer. We deleted this sentence and rewrote as follows: “it was the Edman and Stringer studies at late 80's which demonstrated, through the analysis of their ribosomal RNA, that the Pneumocystis spp. genus belongs to the group of fungi, despite its deficiency of ergosterol in its membrane [7,8]. Figure 1 summarizes the main historical aspects regarding the discovery of Pneumocystis spp.”
- The citation for IWOP-3 is incorrect ref. 9). The correct citation is: Stringer JR and The Pneumocystis Workshop, Revised Nomenclature for Pneumocystis carinii, Euk. Microbiol. 41:5 (121s-122s), 1994.
Answer: The reference 10 was corrected.
- The citation, Frenkel et al. 1999 is not the original publication where Frenkel discerned the difference between the rat vs human It is: Frenkel JK. 1976. Pneumocystis jirovecisp. from man: morphology, physiology, and immunology in relation to pathology. Natl. Cancer Inst. Monogr. 43: 13-27.
Answer: Thank you. We corrected the reference 11.
- Embarking on a review of nomenclature is challenging and the authors are referred to the excellent summary of the problematic past in Redhead et 2006, their reference 11.
Answer: Thanks for the suggestion. We read again the paper, and the paragraph was corrected and the subclass was added: Pneumocystidomycetidae.
- Line 79: canis and P. macacae have not been formally described according to the ICBN, these names are invalid.
Answer: Thanks to the reviewer. The species P. canis and P. macacae were removed from the manuscript. The paragraph was changed as follows: “Other species belonging to the genus of Pneumocystis have been officially named: P. murina, specifically the species that infect mice; P. wakefieldiae and P. carinii (rats); P. oryctolagi (rabbits). Although other Pneumocystis organisms have now been found in different mammals, they are currently named using the trinomial system of special forms (formae speciales)”.
- Line 99: “genetic libraries of mRNA” the reference cited, #14 does not describe such This citation referred to DNA libraries.
Answer: The reference was reviewed again and we corrected the term “mRNA” by “DNA”.
- Line 108: “short” genome is not standard for a reduced genome size.
Answer: Thanks. The term “short” was changed by “reduced”.
- Lines 114-121. Biotrophy was first suggested in the following publication: Cushion MT, Smulian AG, Slaven BE, Sesterhenn T, Arnold J, Staben C, Porollo A, Adamczak R, Meller
- Transcriptome of Pneumocystis carinii during fulminate infection: carbohydrate metabolism and the concept of a compatible parasite. PLoS One. 2007 May 9;2(5):e423. doi: 10.1371/journal.pone.0000423. PMID: 17487271; PMCID: PMC1855432.
Answer: Thanks to the reviewer. We were describing the paper and results found by Cisse OH, therefore, the sentence about biotrophy was misleading and we removed it. For this reason, focused on Cissé et al, main approach as follow: “applying parsimonious models in order to infer the loss and gain of gene families in the evolutionary history of Pneumocystis spp”.
- Lines 127-132. Pneumocystis murina has been used extensively for immunological studies, evaluation of drugs, and many other applications.
Answer: Thanks so much for your suggestion. Three references about the study of Pneumocystis infection in the P. murina model have been reviewed and included (27-29). The following sentence was added: “… while P. murina has been used extensively to study the persistent forms in the lung and immune dynamics of pneumocystis infection in the host”.
- The authors should update their discussion about the by reading recent publications from Hauser, P.
Answer: Thanks so much for your suggestion. Based on your recommendation, we deleted the initial paragraph and eeven articles were reviewed and included (31-37).
We added the following paragraph: “According to available information, it is accepted that the lifecycle of Pneumocystis spp. comprises two main stages: a mononuclear and without cell wall, possibly vegetative one called the trophic form (Previously called trophozoite), and a thick – walled, with multiple nuclear divisions called ascii form (Previously called cystic form) [31]. Observing and quantifying the characteristics of its life cycle has not been easy, in part, because of the lack of a culture that provides the fungus with optimal conditions for reproduction. An asexual phase by binary fission and a possible endogeny carried out by trophic forms has been insufficiently documented previously [32,33], and scientific community is advocating for quantitative studies to understand the life cycle of Pneumocystis spp. [34]. On the other hand, evidence of a sexual phase was first supported by the observation of synaptonemal complexes involved during meiosis in ascii of Pneumocystis spp [35]. In addition, recent comparative genomic analyses have elucidated a unique mating-type locus (MAT) composed by three genes involved in cell differentiation and necessary to initiate the mating process and entry into the sexual cycle, suggesting that the reproductive mechanism of Pneumocystis species is homothallism [36]. The latter means a single cell of Pneumocystis spp. would have the resources necessary to reproduce sexually without the need for a compatible organism. These findings have generated a new discussion of the reproductive cycle of Pneumocystis in the host [34,36,37]”.
- Line 158: What is intended by the word, “homologues”?
Answer: the word "homologues" was changed by “group” .
- Figure 2 is difficult to interpret and could be simplified. Also-what is new about this proposed model? Also- again, what is the evidence for droplet spread?
Answer: With Figure 2 we wanted to summarize the “natural history” using the epidemiological approach described in the answer to comment 1.
We consider that this epidemiological approach allows to understand the different stages of Pneumocystis jirovecii, including the potential transmission. We agree that there are gaps that need to be study, and in the previous and current figure we acknowledge those gaps using question marks.
We also think that having this kind of visual summary allow readers to think about prevention strategies from an epidemiological perspective. As we mentioned previously in comment 1, there is no clear evidence for droplet spread. We redesigned the figure and used the terms described in the answer to comment 1 to avoid misunderstandings.
The new figure 2 is as follow: (Figure can not be shown here, please see the attachment---Assistant editor)
Figura 2. Hypothetical representation of the natural history of Pneumocystis jirovecii. Some of the references that support the definitions used in this figure are: route of transmission [38-41, 57,58], infective form [45], incubation period and period of transmission of a ill patient [55].
Additionally, to improve the understanding of Figure 2, we made an addition of text and a reference
(53) to the paragraph immediately preceding Figure 2, as follows: “However, some important aspects to complete the natural history are unclear: is there direct contact or droplet spread? Independently of the immune status of the host, how many infectious particles (ascii) would be necessary to develop a PcP or to establish a colonization state?”.
- Lines 291-299: the asymptomatic carriage and subsequent reactivation is no longer thought of as a viable source of active infection. The authors need to delve into the literature on this topic.
Answer: We had problems with our English, and we realized that our writing was misleading. We wanted to point out that people can become infected by Pneumocystis jirovecii any thime in their lives, apparently with transitory states of colonization, and that it is unknown the role of asymptomatic carrier on subsequent P. jirovecii pneumonia.
We reviewed again the literature regarding the development of PcP resulting from previous colonization and subsequent reactivation, and we did not find any article that support this sentence.
However, to rule out this hypothesis, it is necessary to have a cohort study with several years of follow-up, taking repeated samples to test over time to demonstrate that the genotype causing the PcP is genetically identical to the colonizing one, and that those lung samples are collected from the ‘whole’ lung of the sick patient to rule out other possible genotypes present in other pulmonary compartments. This kind of study is not feasible due to the high costs that imply this kind of long follow-up.
The paragraph was changed as follows: “The second hypothesis contemplates the asymptomatic acquisition of P. jirovecii at some point in life, which probably happens since childhood and remains in a state of colonization. In fact, it has been suggested that the first contact of the newborn with the fungus occurs at birth, where the mother would be the most likely source of P. jirovecii infection [80,81]”.
- Table 1 needs some kind of organization, perhaps by year of publication. Otherwise, it is hard to follow.
Answer: The table was organized according to year of publication.
Table 1. Documented frequencies of colonization, according to different clinical conditions of the host(Table can not be shown here, please see the attachment---Assistant editor)
We thank you again for your very constructive comments, we learned a lot from these comments. Please let us know of any questions you may have. We greatly appreciate your time and review.
Yours sincerely,
Cristian Vera Marín. MSc.
Corresponding author
Universidad Pontificia Bolivariana cristian.vera.marin@hotmail.com

Reviewer 2 Report
General comments:
The review by Drs. Marin and Valleja focuses on the transmission and colonization of Pneumocystis jirovecii. The review provides the appropriate references and clearly defines the gaps in knowledge. Overall, the review is relevant. The quality of the manuscript will be significantly improved by editing.
The following comments are minor
In addition, other species belonging to this genus were named as follows: P. murina, specifically the species that infects mice; P. wakefieldiae and P.carinii (rats); P. oryctolagi (rabbits); P. canis (dogs) and P. macacae (macacos).
P. canis and P. macacae have not been formally named. P. canis was recently accepted at NCBI but NCBI is not authoritative (see https://www.ncbi.nlm.nih.gov/Taxonomy/Browser/wwwtax.cgi?mode=Info&id=2698477--- but P. macacae is still putative (listed as P. sp ‘macacae’ see - https://www.ncbi.nlm.nih.gov/genome/100573
Lines 100 – 103: “Ma. L. et al. in 2016, also assembled and annotated the genomes of P. jirovecii,” – This statement is misleading: although this paper did provide the genome for P. carinii and P. murina and P. jirovecii, the first P. jirovecii draft genome is from (Cisse, Pagni et al. 2012), first P. carinii data are from (Slaven, Meller et al. 2006). Please rephrase the sentence to provide proper credits
Lines 108-110: “however, compared to the genomes of S. pombe and T. deformans, it is a short genome with a limited set of genes, which could explain the dependence and adaptation of P. jirovecii to the human host [17]” – this statement is purposely vague and non-informative. Please be specific and explain which compounds are missing and how exactly this related Pneumocystis host adaption (please refer to the following reviews and the reference therein for nutritional dependencies (Cisse and Hauser 2018, Ma, Cisse et al. 2018)
Line 130: “P. carinii” should be italicized
Lines 133 to 138: The reference 14 is not appropriate here. There is no definitive proof that Pneumocystis trophs divide by binary fission. In fact, the possibility of Pneumocystis being strictly sexual – no asexual cycle -- is currently debated (see (Hauser PM 2018, Hauser 2021)
Line 39: Can you please expand on the reference 24 is used. This is a metagenomics study that explore the hypothesis of Pneumocystis airborne transmission in the vicinity of healthy individuals.
Line 147: “P. carinii” should be italicized
Line 151: please provide the reference just after “in 2010 by Cushion et al.”
Figure 2: The term “por” should be replaced. Overall, the text contains multiple Spanish words that should be carefully edited
Lines 285 – 287: The ref 62 by Nevez et al (https://pubmed.ncbi.nlm.nih.gov/16652305/), which reports an absence of P. jirovecii in healthy subjects, seems incompatible with this statement. Although the focus on this study is P. jirovecii, animal models of P. carinii demonstrate that the healthy hosts are important for transmission (for example (Gigliotti, Harmsen et al. 2003). In fact, the authors might want to include animal models in this review
Lines 310: Please refrain from mixing the old name “P. carinii f. sp. Hominis” which can be misleading. If unavoidable, please clearly states this nomenclature is only used for historical purposes and is not used anymore.
Lines 352 and 397: “P. jirovecii” should be italicized
References
Cisse, O. H. and P. M. Hauser (2018). "Genomics and evolution of Pneumocystis species." Infect Genet Evol 65: 308-320.
Cisse, O. H., L. Ma, J. P. Dekker, P. P. Khil, J. H. Youn, J. M. Brenchley, R. Blair, B. Pahar, M. Chabe, K. K. A. Van Rompay, R. Keesler, A. Sukura, V. Hirsch, G. Kutty, Y. Q. Liu, L. Peng, J. Chen, J. Song, C. Weissenbacher-Lang, J. Xu, N. S. Upham, J. E. Stajich, C. A. Cuomo, M. T. Cushion and J. A. Kovacs (2021). "Genomic insights into the host specific adaptation of the Pneumocystis genus." Communications Biology 4(1).
Cisse, O. H., M. Pagni and P. M. Hauser (2012). "De novo assembly of the Pneumocystis jirovecii genome from a single bronchoalveolar lavage fluid specimen from a patient." MBio 4(1): e00428-00412.
Gigliotti, F., A. G. Harmsen and T. W. Wright (2003). "Characterization of transmission of Pneumocystis carinii f. sp. muris through immunocompetent BALB/c mice." Infect Immun 71(7): 3852-3856.
Hauser, P. M. (2021). "Pneumocystis Mating-Type Locus and Sexual Cycle during Infection." Microbiol Mol Biol Rev 85(3): e0000921.
Hauser PM, C. M. (2018). "Is sex necessary for the proliferation and transmission of Pneumocystis?" PLOS Pathogens 14(12): e1007409.
Ma, L., O. H. Cisse and J. A. Kovacs (2018). "A Molecular Window into the Biology and Epidemiology of Pneumocystis spp." Clin Microbiol Rev 31(3): e00009-00018.
Slaven, B. E., J. Meller, A. Porollo, T. Sesterhenn, A. G. Smulian and M. T. Cushion (2006). "Draft assembly and annotation of the Pneumocystis carinii genome." J Eukaryot Microbiol 53 Suppl 1: S89-91.
Author Response
Dear Editor,
We want to thank for the detailed and constructive review, including the amazing references and clear suggestions and feedback that the two reviewers did about our paper. We feel priviledge that experts in this field agreed to review our paper and we think this is one of the most educational feedback that we have ever received. We address all your comments and they were very helpful to improve our paper dramatically. We also want to thank that both reviewers gave us the opportunity to improve our paper despite our English mistakes.
We paid the review of the English by a native English speaker, but we haven’t received yet that file. As we already requested one extension because we were waiting for this version, we decided to submit the paper with our changes and corrections to meet the deadline, but we clarify that we are still waiting for the review of our English.
Please find the answers to each comment or question below. Based on your petition, we send to the
Journal of Fungi the article with the changes made "Track changes", so that it can be easily identified.
Reviewer(s)' Comments to Author: Referee: 2
- In addition, other species belonging to this genus were named as follows: P. murina, specifically the species that infects mice; P. wakefieldiae and P.carinii (rats); P. oryctolagi (rabbits); canis (dogs) and P. macacae (macacos). P. canis and P. macacae have not been formally named. P. canis was recently accepted at NCBI but NCBI is not authoritative (see https://www.ncbi.nlm.nih.gov/Taxonomy/Browser/wwwtax.cgi? mode=Info&id=2698477--- but P. macacae is still putative (listed as P. sp ‘macacae’ see - https://www.ncbi.nlm.nih.gov/genome/100573
Answer: Thanks to the reviewers. The paragraph has been modified as follows: “Other species belonging to the genus of Pneumocystis have been officially named: P. murina, specifically the species that infect mice; P. wakefieldiae and P. carinii (rats); P. oryctolagi (rabbits).
Although other Pneumocystis organisms have now been found in different mammals, they are currently named using the trinomial system of special forms (formae speciales)”. The species P. canis and P. macacae were removed from the manuscript.
- Lines 100 – 103: “Ma. L. et al. in 2016, also assembled and annotated the genomes of P. jirovecii,” – This statement is misleading: although this paper did provide the genome for
- carinii and P. murina and P. jirovecii, the first P. jirovecii draft genome is from (Cisse, Pagni et al. 2012), first P. carinii data are from (Slaven, Meller et al. 2006). Please rephrase the sentence to provide proper credits.
Answer: We are grateful for the comments and recommended references. The articles were reviewed and included in the manuscript. The sentence was changed as follows: “This project resulted in the first sequenced and annotated draft of the P. carinii genome [16], and subsequently, in 2012, the genome of P. jirovecii was documented for the first time, reconstructed with DNA from four bronchoalveolar lavage (BAL) samples from patients with PcP using cell immunoprecipitation and random DNA amplification techniques. The analysis by Cissé et al. documented important genomic characteristics of P. jirovecii, including a genome length of 8.1 Mb, a low GC content (29%), and a striking absence of enzymes related to the synthesis of virulence factors, toxins and enzymes involved in metabolic amino acid pathways, the latter being a characteristic of obligate parasites [17]”.
In addition, the references recommended by the reviewer were included in the paper: “16. Slaven BE, Meller J, Porollo A, Sesterhenn T, Smulian AG, Cushion MT. Draft Assembly and Annotation of the Pneumocystis carinii Genome. J Eukaryot Microbiol. 2006;53(s1):S89-91”; and “17. Cissé OH, Pagni M, Hauser PM. De novo assembly of the Pneumocystis jirovecii genome from a single bronchoalveolar lavage fluid specimen from a patient. mBio. 26 de diciembre de 2012;4(1):e00428-00412”.
- Lines 108-110: “however, compared to the genomes of S. pombe and T. deformans, it is a short genome with a limited set of genes, which could explain the dependence and adaptation of P. jirovecii to the human host [17]” – this statement is purposely vague and non- Please be specific and explain which compounds are missing and how exactly this related Pneumocystis host adaption (please refer to the following reviews and the reference therein for nutritional dependencies (Cisse and Hauser 2018, Ma, Cisse et al. 2018).
Answer: Thanks to the recommendation of these references, the documents were reviewed and included in the manuscript (references 21 and 22).
The paragraph was modified and complemented to include the genomic characteristics and metabolic pathways lost or absent in Pneumocystis species, as follows: “… compared to the 5,044 genes obtained on average from the Schizosaccharomyces genomes analyzed. In this analysis, the gene family with the most significant reduction was observed in the Pfam domains, evidenced by the low number of transporter proteins, transcription factors and enzymes (oxidoreductases, hydrolases, transferases and coenzymes). In addition, other metabolic pathway-related capabilities of the fungus were also found to be reduced and/or absent in all three Pneumocystis species analyzed, including
- jirovecii, e.g., loss of pathways involved in amino acid synthesis, (results consistent with those obtained by Cissé in 2006, [17]), and a reduced capacity for nitrogen and sulfur assimilation, lipid and carbohydrate metabolism, glycerol synthesis and cofactor metabolism. On the other hand, Ma.
- et al. also documented enriched protein domains on the cell surface belonging to the Msg superfamily, which have important functions related to the interaction with the host and antigenic variation. These findings could explain the dependence and adaptation of P. jirovecii to the human host [20,21]. Likewise, later work by Cissé et. al. documented that the adaptation of Pneumocystis spp. to the host could be explained, in part, by the highly polymorphic multicopy gene families (msgs), which evolutionarily have provided the fungus with antigenic variation capabilities and necessary mechanisms for immune evasion within the host [22].”
- Line 130: “P. carinii” should be italicized.
Answer: The genus and species of the organism were italicized.
- Lines 133 to 138: The reference 14 is not appropriate here. There is no definitive proof that Pneumocystis trophs divide by binary fission. In fact, the possibility of Pneumocystis being strictly sexual – no asexual cycle -- is currently debated (see (Hauser PM 2018, Hauser 2021).
Answer: Thanks for your suggestion. We made a mistake inserting the reference 24. We rewrote the paragraph according to the recommended reading of the most current articles discussing the subject of Pneumocystis reproduction in the host, written by Dr. Philippe Hauser and collaborators.
Finally, the paragraph was modified as follows: “According to available information, it is accepted that the lifecycle of Pneumocystis spp. comprises two main stages: a mononuclear and without cell wall, possibly vegetative one called the trophic form (Previously called trophozoite), and a thick – walled, with multiple nuclear divisions called ascii form (Previously called cystic form) [31]. Observing and quantifying the characteristics of its life cycle has not been easy, in part, because of the lack of a culture that provides the fungus with optimal conditions for reproduction. An asexual phase by binary fission and a possible endogeny carried out by trophic forms has been insufficiently documented previously [32,33], and scientific community is advocating for quantitative studies to understand the life cycle of Pneumocystis spp. [34]. On the other hand, evidence of a sexual phase was first supported by the observation of synaptonemal complexes involved during meiosis in ascii of Pneumocystis spp [35]. In addition, recent comparative genomic analyses have elucidated a unique mating-type locus (MAT) composed by three genes involved in cell differentiation and necessary to initiate the mating process and entry into the sexual cycle, suggesting that the reproductive mechanism of Pneumocystis species is homothallism [36]. The latter means a single cell of Pneumocystis spp. would have the resources necessary to reproduce sexually without the need for a compatible organism. These findings have generated a new discussion of the reproductive cycle of Pneumocystis in the host [34,36,37]”.
6. Line 39: Can you please expand on the reference 24 is used. This is a metagenomics study that explore the hypothesis of Pneumocystis airborne transmission in the vicinity of healthy individuals.
Answer: The article was included in the manuscript with the following description: “Cissé O. et al. documented in 2020 an analysis based on metagenomic techniques to explore the dynamics of human environmental exposure to Pneumocystis spp. This analysis was derived from an exposome study [56], where they set out to develop a method for personalized monitoring and tracking of biological and chemical exposure in air; the monitoring included 15 adult participants defined as healthy, who were fitted with a personal exposure monitor for the collection of biotic and abiotic samples for aerosol capture, with a follow-up period of 890 days at 66 different geographic locations. With the reads obtained and to obtain genetic traces of P. jirovecii, the data were compared with reference genomic information available at NCBI, as well as for the detection of DNA from different mammals. The authors found evidence of Pneumocystis spp. in 24/594 SRA (sequence read archive) files, which represent samples obtained from 4/15 study participants. De novo assembly of the 24 SRA files resulted in 45 contigs with more than 500 nucleotides, of which 37/45 contigs were identified as P. jirovecii. Given the host specificity of Pneumocystis species, and their inability to reproduce outside the host, these results provide evidence of short-range aerosol exposure and possible transmission of jirovecii, and the role of healthy individuals as potential transmitters. Thus, metagenomic methods provide important opportunities for understanding the transmission dynamics of pathogens, in this case, P. jirovecii.
- Line 147: “P. carinii” should be italicized.
Answer: The genus and species of the organism was italicized.
- Line 151: please provide the reference just after “in 2010 by Cushion et al”.
Answer: The reference was added where the reviewer suggested.
- Figure 2: The term “por” should be replaced. Overall, the text contains multiple Spanish words that should be carefully edited.
Answer: The word "por" was replaced by "by" in Figure 2. In general, the entire manuscript in under review by an English native speaker.
- Lines 285 – 287: The ref 62 by Nevez et (https://pubmed.ncbi.nlm.nih.gov/16652305/), which reports an absence of jirovecii in healthy subjects, seems incompatible with this statement. Although the focus on this study is P. jirovecii, animal models of P. carinii demonstrate that the healthy hosts are important for transmission (for example (Gigliotti, Harmsen et al. 2003). In fact, the authors might want to include animal models in this review.
Answer: We agree with the reviewer's statement “animal models of P. carinii demonstrate that the healthy hosts are important for transmission”. We reviewed the studies of Gigliotti F and we have included the reference https://doi.org/10.1371/journal.ppat.1003025 to support the route of transmission to the host in Figure 2 (see reference 57).
The message we wanted to describe in the section: Colonization of P. jirovecii in different population groups is the wide range of colonization frequencies that have been documented specifically in the species P. jirovecii in the human host, highlighting the different population types and clinical conditions in which the fungus has been detected.
We seriously contemplated and reviewed animal model studies to include in this section, however, we consider that it is beyond the scope of this section of the review.
- Lines 310: Please refrain from mixing the old name “P. carinii f. sp. Hominis” which can be misleading. If unavoidable, please clearly states this nomenclature is only used for historical purposes and is not used
Answer: Thanks for the suggestion. The old name "P. carinii f. sp. Hominis" is replaced by "P. jirovecii". We reviewed the entire manuscript to avoid misleading, except in line 75, which it is given from a historical perspective.
- Lines 352 and 397: “P. jirovecii” should be italicized Answer: The genus and species of the organism was
We thank you again for your very constructive comments, we learned a lot from these comments. Please let us know of any questions you may have. We greatly appreciate your time and review.
Yours sincerely,
Cristian Vera Marín. MSc.
Corresponding author
Universidad Pontificia Bolivariana cristian.vera.marin@hotmail.com

Round 2
Reviewer 1 Report
Please see file. Corrections must be made before publication.

Author Response
Medellín, November 1, 2021
Editor
Journal of Fungi
Dear Editor,
We thank again to the reviewer 1 for her/his comments, we sincerely appreciate all of them. The new version has the English proofreading that was requested in the round 1 of both reviewers. The English review was made a Native English Professor who corrected the grammar and typos. It also has the corrections suggested by the reviewer 1 in round 2.
We attach the file with track changes. Please find the answers to each comment or question below.
Reviewer(s)' Comments to Author:
Referee: 1
- Extensive editing of English language and style required.
Answer: According to the reviewer's recommendation, the entire manuscript was reviewed by a Native English Professor who corrected the grammar and typos.
- Please see file. Corrections must be made before publication.
Answer: Thank you so much for your corrections, we included in the paper.
- “Asci” is misspelled throughout (not ascii).
Answer: Thanks, we corrected in the paper and the figure.
- Line 38: Antonio should be spelled out
Answer: It was incorporated within the paper.
- Line 79: canis and P. macacae have not been formally described according to the ICBN, these names are invalid. Add at the end of the paragraph “or other invalid systems”
Answer: Thanks to the reviewer. The species P. canis and P. macacae were removed from the manuscript. The paragraph was changed as follows: “Other species belonging to the genus of Pneumocystis have been officially named: P. murina, specifically the species that infect mice; P. wakefieldiae and P. carinii (rats); P. oryctolagi (rabbits). Although other Pneumocystis organismis have now been found in different mammals, they are currently named using the trinomial system of special forms (formae speciales)” or other invalid systems.
- Lines 116-120: With the findings in the species P. jirovecii, we now know that the genome of these species 116 has a length of approximately 8.4 Mb distributed in 20 chromosomes, a GC content percentage of 28.4% and approximately 3,761 protein-coding genes, however, compared to the genomes of S. pombe and T. deformans, compared to the 5,044 genes obtained on average from the Schizosaccharomyces genomes analyzed. This sentence does not make sense.
Answer: We deleted the underline sentence that was confusing.
- Please see the following corrections to this new paragraph lines 158-175.
Answer: Thank you very much, we incorporated the corrections made by the reviewer within the article.
- Figure 2. This figure remains confusing and needs a more detailed legend. The use of pre-pathogenic and pathogenic periods is questioned: https://www.cdc.gov/csels/dsepd/ss1978/lesson1/section9.html The drawings meant to indicate asci (one “i” please) look like the single-celled trophic forms. This should be changed.
Answer: Thank you very much! We corrected that those two terms and used the definitions of CDC: Stage of susceptibility, of clinical disease, clinical disease and clinical outcomes. We also corrected the asci typo, and we modified the shape. Finally, we included an explanation to the figure.
- Lines 188-197 are incorrect. Please note the corrections.
Answer: Thank you very much, we really appreciate the corrections made by the reviewer. We incorporated them within the article.
We thank again for your suggestions, our paper improved dramatically thanks to the generous feedback of the reviewers.
Yours sincerely,
Both authors
